# Influence of Residual Feed Intake and Cow Age on Dry Matter Intake Post-Weaning and Peak Lactation of Black Angus Cows

**DOI:** 10.3390/ani11061822

**Published:** 2021-06-18

**Authors:** Cory T. Parsons, Julia M. Dafoe, Samuel A. Wyffels, Timothy DelCurto, Darrin L. Boss

**Affiliations:** 1Northern Agricultural Research Center, Montana State University, Havre, MT 59501, USA; julia.dafoe@montana.edu (J.M.D.); samwyffels@montana.edu (S.A.W.); dboss@montana.edu (D.L.B.); 2Department of Animal and Range Sciences, Montana State University, Bozeman, MT 59717, USA; timothy.delcurto@montana.edu

**Keywords:** beef cattle, cow age, dry matter intake, residual feed intake, stage of production

## Abstract

**Simple Summary:**

Supplemental nutrition for cattle is the greatest operating cost for cow-calf producers, accounting for 65% of the annual expenses. In addition, residual feed intake (RFI) is being used as a selection tool for purchasing and retaining heifers, as well as selecting bulls with the goal of improving feed efficiency and/or reducing supplemental inputs. However, the use and relevance of RFI as a selection tool for the cow-calf industry needs additional research. In our studies, heifer post-weaning RFI did not influence mature cow dry matter intake and intake behavior for both lactating and non-lactating beef cows. In contrast, cow age did correspond to increases of intake and intake rates of mature cows. However, when intake was expressed as g/kg body weight^−1^, no differences were observed with respect to cow age for lactating and non-lactating cows. Milk production was influenced by heifer post-weaning RFI in 5–6- and 8–9-year-old cows, however, did not influence 9–10-year-old cows. Therefore, our research suggests that cow age has greater impacts on dry matter intake than RFI, however, the relationship between RFI of heifers and subsequent mature cow milk production warrants further investigation.

**Abstract:**

We evaluated heifer post-weaning residual feed intake (RFI) classification and cow age on dry matter intake (DMI) at two stages of production. Fifty-nine non-lactating, pregnant, (Study 1) and fifty-four lactating, non-pregnant (Study 2) commercial black Angus beef cows were grouped by age and RFI. Free-choice, hay pellets were fed in a GrowSafe feeding system. In Study 1, cow DMI (kg/d) and intake rate (g/min) displayed a cow age effect (*p* < 0.01) with an increase in DMI and intake rate with increasing cow age. In Study 2, cow DMI (kg/d) and intake rate (g/min) displayed a cow age effect (*p* < 0.02) with an increase in DMI and intake rate with increasing cow age. Milk production displayed a cow age × RFI interaction (*p* < 0.01) where both 5–6-year-old and 8–9-year-old low RFI cows produced more milk than high RFI cows. For both studies, intake and intake behavior were not influenced by RFI (*p* ≥ 0.16) or cow age × RFI interaction (*p* ≥ 0.21). In summary, heifer’s post-weaning RFI had minimal effects on beef cattle DMI or intake behavior, however, some differences were observed in milk production.

## 1. Introduction

Supplemental nutrition for cattle is the greatest operating cost for cow-calf producers, accounting for 65% of the annual expenses [1,2,3]. Traditionally, selection pressure has been placed on production traits associated with increasing outputs (average daily gain, weaning weight, yearling weight, etc.), which can also result in increased inputs to achieve animal production potential. Since feed costs constitute the greatest proportion of total inputs, selection pressure for efficient animals that have lower feed intake but maintain production, or average intake with higher production, could have a great impact on cow-calf profitability [3]. Thus, improving feed efficiency through genetic selection holds significant opportunity for the beef industry.

Residual feed intake (RFI) is currently being used as a selection tool for purchasing and retaining heifers and for selecting bulls. Most studies have used steers and terminal heifers when evaluating RFI impact on various aspects of beef cattle production [4,5,6]. However, the use and relevance of RFI as a selection tool for the cow-calf industry needs additional research [7,8,9]. Most RFI studies have included energy-dense diets and rations focusing on feedlot performance [4,6,10]. Research pertaining to RFI of cattle offered forage-based diets is limited [11], with even less data available related to beef cattle forage-based production systems [3,7,12,13]. As a result, more research is needed to evaluate the utility of RFI estimates on beef cattle production systems in extensive forage-based environments [8,12,14].

The use of RFI for beef cattle selection in rangeland environments is based on the assumption that heifer post-weaning RFI will be expressed throughout the lifetime productivity of that heifer [9]. Past research suggests that RFI is a moderately heritable trait [15], however, few published studies exist comparing RFI of individuals of two different physiological states [5,6,16]. Multiple studies have investigated feed efficiency in young, pre-mature cattle, but few have quantified how RFI relates to feed efficiency at different ages and stages of production [16,17,18]. The repeatability of RFI between growing and finishing phases of beef cattle production has recently been examined [6,19], but there are few reports demonstrating the relationship of RFI of growing females compared with those same females as mature, lactating [17,20], and non-pregnant, non-lactating beef cows [18].

Research evaluating heifer post-weaning RFI on subsequent cow intake and intake efficiency is limited. Therefore, the objectives of these research studies were to: 1) evaluate the effects of heifer post-weaning RFI on non-lactating, pregnant, beef cows dry matter intake (DMI) at different cow ages; and 2) evaluate the effects of heifer post-weaning RFI on lactating, non-pregnant, beef cow DMI at different cow ages, as well as peak milk production. We hypothesized that heifers identified as low RFI eat less than high RFI cows and the influence of RFI may interact with cow age.

## 2. Materials and Methods

The use of animals in this study was approved by the Agricultural Animal Care and Use Committee of Montana State University AACUC #2018-AA12. These studies were conducted at the Northern Agricultural Research Center (NARC), located in Havre, Montana. All cattle were synchronized and time artificially inseminated (AI) corresponding to initial calving date of March 15, with exposure to cleanup bulls occurring approximately 7 days post AI for an additional 45 days. Calves were strategically weaned mid-September to mid-October each year with the timing based on fall forage conditions and/or cow body condition.

All cows were managed as one contemporary group post-weaning in the fall to calving in the spring of each year. All heifers/cows were exposed to electronic feed bunks as heifers and again during winter supplement studies.

### 2.1. Heifer RFI Trials

Starting in 2011, all Northern Agricultural Research Center cattle have been utilized in a heifer RFI trial for a minimum of 77 days on a forage-based ration provided in a GrowSafe system (GrowSafe DAQ 4000E; GrowSafe System Ltd., Airdrie, AB, Canada). Calves were weaned on pasture mid-September to early October each year and entered an RFI trial 60 to 75 days post weaning. On average, 45–95 AI sired replacement heifers were retained annually. Upon RFI trial initiation, and completion, all heifers were weighed post feeding, on two consecutive days to record beginning and ending body weights (BW), then again, every 28 days to record BW gain. A 7-d acclimation period preceded a 70-d feeding trial, while the GrowSafe system recorded individual daily feed intakes. All heifers had free access to 16–32 GrowSafe feed bunks (depending on year and number of heifers) and ad libitum access to water and forage-based diets, consisting of 30.4% corn silage, 41.1% grass hay, and 28.5% alfalfa on a dry mater basis, formulated to meet requirements for growing moderate frame beef heifers (10.5% CP and 66.0% TDN [21]). Individual heifer post-weaning RFI was calculated following parameters set forth by Archer et al. [22] and Arthur et al. [4]:RFI_Phe_ = FI-β w(_phe_) × MWT-βg(_phe_) × DG,(1)
where RFI_phe_ = phenotypic residual feed intake, FI = daily feed intake, MWT = metabolic body weight at mid-test, DG = average daily gain, and βw(_phe_) and βg(_phe_) = partial regression coefficients of animal’s FI on MWT and DG, respectively. Heifers were classified as either low (>−0.50 SD from mean), or high (<+0.50 SD from the mean) within a year [9].

Two feed intake studies were conducted to evaluate the impacts of heifer post-weaning RFI at differing ages and stages of beef cattle production. Individual BW and body condition scores (BCS) were recorded and cattle were placed in a GrowSafe feeding system for DMI analysis over a 21-d period. For each study, treatments were replicated in two pens, each containing 16 GrowSafe feeding units. Individual animal intake was continuously recorded. The system was monitored daily for unaccounted feed balance. When greater than 5% of the feed disappearance was unaccounted for, the GrowSafe system automatically deemed the 24-h period as failed. Therefore, we selected the last 7 days of the 21-d period that met the criteria to calculate average DMI per individual animal for each DMI study. Variation in dry matter intake (kg/d), measured as the coefficient of variation (% CV), was based on daily intake estimates for individual animals.

### 2.2. Study 1: Non-Lactating, Pregnant Cow

Fifty-nine non-lactating, pregnant, black Angus females were utilized to evaluate the impacts of heifer post-weaning RFI on DMI post weaning. All cows were fed commercially available free-choice alfalfa/straw pellet formulated to meet nutrient requirements for non-lactating, pregnant cows (Table 1; CHS Nutrition, Sioux Falls, SD, USA; [21]). At the initiation of the trial (14-d post weaning), cows were classified by age, (1–2, 4–5, and 7–8 years old;) and within age class represented both low and high RFI, and dry-lotted for 16 h to obtain uniform shrunk BW and BCS.

### 2.3. Study 2: Lactating, Non-Pregnant Cow

Fifty-four lactating, non-pregnant, black Angus females were utilized to evaluate the impacts of heifer post-weaning RFI (low and high) and cow age (2–3-, 5–6-, and 8–9-year-old cows) and dry-lotted for 16 h to obtain uniform shrunk BW and BCS. Cows were selected by the same age and RFI criteria as described for Study 1. However, for this study, we only utilized cows that calved within the first 42 days of the calving period. Calf BW and Julian birth date were measured to characterize the influence of cow age and RFI. All cows were fed commercially available free-choice alfalfa/grass base pellet formulated to meet NRC requirements for lactating cows for the DMI study (Table 1; CHS Nutrition, Sioux Falls, SD, USA; [21]). At the conclusion of the 7-d DMI period, approximately day 60 post calving, a weigh-suckle-weigh trial was conducted to evaluate the impacts of heifer post-weaning RFI and cow age on milk production following methods detailed by Williams et al. [23]. For our study, we utilized an 8-h calf removal protocol.

### 2.4. Statistical Analysis

For both Studies 1 (Appendix A Appendix A) and 2 (Appendix A Appendix A), the influence of RFI and cow age on initial cow BCS and BW were analyzed using ANOVA with a generalized linear model including RFI, cow age and the interactions of RFI and cow age as fixed effects. Additionally, the influence of RFI and cow age on intake and intake behavior were analyzed using ANOVA with a generalized linear mixed model including RFI, cow age, and the interactions of RFI and cow age as fixed effects, and individual cow and pen as random effects. When RFI × cow age interactions were observed means were separated within age groups. Individual animal was considered the experimental unit and an alpha ≤ 0.05 was considered significant. Orthogonal polynomial contrasts were used to determine linear and quadratic effects for cow age. Means were separated using the Tukey method when *p* < 0.05. Tendencies were reported when significance was *p* ≤ 0.10. All statistical analyses were performed in R [24].

## 3. Results

### 3.1. Study 1: Non-Lactating, Pregnant Cow

Cow BW displayed a cow age × RFI interaction (*p* < 0.01) where 4–5-year-old low RFI cows had a lighter BW than high RFI cows, however, low RFI 7–8-year-old cows tended (*p* = 0.06) to have greater BW than high RFI cows (Table 2). No differences in BW were observed between RFI classifications in 1–2-year-old cows (*p* = 0.17). Cow BCS also displayed a cow age × RFI interaction (*p* = 0.02) where low RFI 4–5-year-old cows had lower BCS than high RFI 4–5-year-old cows with no differences observed between RFI in other cow ages (*p* ≥ 0.24). Both DMI (kg/d) and intake rate (g/min) displayed a cow age effect (*p* < 0.01) with a quadratic increase with increasing cow age (*p* ≤ 0.02; Table 2). Specifically, young cows (1–2-year-old) ate less and had lower intake rates than older cows (*p* < 0.01). Neither DMI (g/kg of BW), intake variation (% CV), or time spent at the feeder (min/d) were affected by cow age or RFI (*p* ≥ 0.16), averaging 28.8 g/kg of BW, 19.0% CV, and 107.5 min/d, respectively.

### 3.2. Study 2: Lactating, Non-Pregnant Cow

Cow BW displayed a cow age × RFI interaction (*p* < 0.01), with low RFI 5–6- and 8–9-year-old cows having a greater BW than high RFI 5–6- and 8–9-year-old cows (*p* ≤ 0.01; Table 3). Similar to cow BW, cow BCS displayed a cow age × RFI interaction (*p* < 0.01) where low RFI 5–6-year-old cows had lower BCS than high RFI 5–6-year-old cows (*p* < 0.01), whereas, low RFI 8–9-year-old cows had higher BCS than low RFI 8–9-year-old cows (*p* < 0.01).

Calf BW, measured at weigh-suckle-weigh, displayed a cow age × RFI interaction (*p* <0.03), where calf BW from 2–3-year-old cows were lower in high RFI 2–3-year-old cows compared to low RFI cows (*p* = 0.02; Table 3). Calf Julian birth date was affected by cow age (*p* < 0.01) with calf birth date increasing linearly with increasing cow age (*p* < 0.01; Table 3). There was also a tendency for an RFI effect on calf Julian birth date (*p* = 0.09) with low RFI cows tending to calve later in the calving season than high RFI cows, averaging 72.2 and 73.9 for high and low RFI, respectively.

Cow DMI (kg/d) displayed a cow age effect (*p* < 0.01) with a quadratic increase in DMI with increasing cow age (*p* < 0.01; Table 3). Similarly, intake rate (g/min) displayed a cow age effect (*p* < 0.02) with a linear increase (*p* < 0.01) in intake rate with increasing cow age. In contrast, neither DMI intake (g/kg of BW), % CV, nor time spent at the feeder (min/d) were affected by cow age, or RFI, averaging 21.7 g/kg of BW, 11.7% CV, and 149.7 min/d, respectively.

Cow milk production (kg) displayed a cow age × RFI interaction (*p* < 0.01; Table 3), with 2–3-year-old and 5–6-year-old low RFI cows producing more milk than high RFI cows of the same age (*p* < 0.01). In addition, cow milk production (g/kg of BW), displayed a cow age × RFI interaction (*p* < 0.01) with 2–3-year-old and 5–6-year old low RFI cows producing more milk per kg of BW than high RFI cows of the same age (*p* < 0.01), however, no differences were observed in 8–9-year-old cows (*p* = 0.48).

## 4. Discussion

It has been reported that as cattle grow and mature, composition of their gain changes from protein accretion to fat deposition [25]. Since the expense of protein accrual is less than for fat deposition, the efficiency that cattle convert feed into BW gain is reduced as they mature [26]. Multiple research papers have reported on the changes in feed efficiency (RFI) of cattle at different stages of physiological growth. Durunna et al. [19] reported that following two consecutive 70-d RFI periods, 49% of the heifers maintained their original RFI classification, whereas 51% had a different RFI classification, indicating re-ranking exists in heifers despite receiving the same basal diet. Loyd et al. [16] suggested that RFI determined during the pre-pubertal period may only be a moderate predictor of post-pubertal RFI.

Archer et al. [22] reported a moderate correlation of 0.40 between RFI measured in heifers post-weaning and later as non-gestating, non-lactating 3-year-old cows. Freetly et al. [18] compared the RFI classification of yearling heifers following an 84-d RFI intake trial with subsequent RFI classification of 5-year-old non-pregnant, non-lactating cows, and reported that feed intake and ADG are heritable and genetically correlated between heifers and cows. Black et al. [17] compared the RFI classification of growing heifers following a 70-d RFI trial and subsequently as 3-year-old lactating beef cows and reported heifers that were the most feed efficient consumed less feed as lactating cows while maintaining similar performance. However, they reported that correlations between heifer and mature cow RFI values were not significant indicating that within-animal feed efficiency was not maintained as the calves developed from growing heifers to mature, lactating cows.

Results from our research are in general agreement with previous research where intake tends to change between ages and physiological stages of production of female beef cows, indicating that heifer post-weaning RFI may not be a reliable predictor of mature cow feed intake. In contrast, cow age had a substantial impact on intake, and intake behavior. Previous research reported that cow age significantly impacted grazing behavior and terrain use when comparing older cows to younger cows [9,27,28]. Furthermore, although limited, previous studies evaluating supplement intake of mixed-age beef herds have reported that younger cows spent less time at feeders and consumed less feed than older cows [29,30]. In contrast, Wyffels et al. [28] reported younger cattle consumed more and visited the feeders more often than older cows. Parsons et al. [13] observed a quadratic effect in intake related to cow age where 1-year-old cows consumed more supplement and had a larger CV of intake than older cows.

Broleze et al. [20] states that measuring RFI in lactating animals is important, however, RFI does not accurately reflect production efficiency. This is because the RFI models that are used to calculate residual traits for lactating cows do not account for the energy partitioning into the various components, some of which are more financially important (e.g., milk fat and protein yield) than others (metabolic BW; [31]). Previous research found no relationship between milk yield (obtained by weigh-suckle-weigh technique) and RFI, with low RFI and high RFI cows having similar milk yield [10,17,32]. Rutledge et al. [33] reported that dams nursing female calves produced significantly more milk than those nursing male calves. In contrast, Melton et al. [34] and Christian et al. [35] found no significant difference in dam’s milk yield attributable to sex of calf while Pope et al. [36] reported an advantage for cows nursing male calves.

In this study, cow milk production (g/kg of BW^−1^) was impacted by both cow age and RFI with 3–4- and 6–7-year-old low RFI cows producing more milk than high RFI cows of the same age. Regardless of unit of measurement, the difference in milk production (kg vs. g/kg of BW) were consistent across young and middle-age cows. This suggests that heifer post-weaning RFI may be related to milk production of young and middle-aged cows. Few research trials have compared low RFI to high RFI lactating, non-pregnant beef cows [10,17,37]. Previous research report that low RFI (more efficient) and high RFI (less efficient) cows produce similar quantities of milk, but the former consumed less dry matter per day [17,32,38]. To our knowledge, this is the first study comparing post-weaning heifer RFI to subsequent milk production of beef cows across multiple age classes.

## 5. Conclusions

Heifer post-weaning RFI did not influence mature cow dry matter intake, and this was consistent for both lactating and non-lactating beef cows. In contrast, cow age did correspond to quadratic increases of DMI and intake rates of mature cows. However, when DMI was expressed as g/kg of BW no differences were observed with respect to cow age in lactating and non-lactating cows. Milk production was influenced by heifer post-weaning RFI for 3–4- and 6–7-year-old cows, however, did not influence 9–10-year-old cows. Therefore, our research suggests that cow age has greater impacts on dry matter intake than RFI, however, the relationship between RFI of heifers and subsequent mature cow milk production warrants further investigation.

## Figures and Tables

**Table 1 animals-11-01822-t001:** Ingredients and nutrient composition (DM basis) of the commercially available hay pellets provided ad libitum to cows in GrowSafe pens during the 21-d DMI trials, fall 2019 and spring 2020.

Item	Non-Lactating, Pregnant Cow	Lactating, Non-Pregnant Cow
Ingredient, %	-	-
Alfalfa hay	49.53	79.05
Straw	49.52	-
Corn, ground	-	20.0
Ultramin 12-6	0.75	0.75
Trace mineral mix ^1^	0.20	0.20
Nutrient value, %	-	-
Dry Matter	93.6	90.4
Crude Protein	10.5	16.8
Acid Detergent Fiber	40.4	34.0
Total Digestible Nutrients	62.0	65.1
Net Energy maintenance	0.63	0.67
Net Energy gain	0.36	0.40
Net Energy lactation	0.64	0.67
Ca	0.96	1.79
P	0.19	0.21
S	0.19	0.21
Mg	0.20	0.27
Na	0.02	0.02
K	2.03	2.03

^1^ Trace mineral mix: 285.5 ppm Fe, 71.0 ppm Zn, 59.0 ppm Mn, 25.0 ppm Cu, 1.6 ppm I, 1.45 ppm Mo, 0.4 ppm Se, 1.0 IU/kg vitamin A, 0.1 IU/kg vitamin D, 1.7 IU/kg vitamin E.

**Table 2 animals-11-01822-t002:** The influence of heifer post-weaning RFI on subsequent beef cow performance and dry matter intake behavior of 3 age classes of non-lactating, pregnant beef cows (Study 1).

Category	Cow Age, Years						-	*p*-Value		
	1/2		4/5		7/8					
	Low RFI	High RFI	Low RFI	High RFI	Low RFI	High RFI	SE ^1^	Age	RFI	Age × RFI
Cows *n*	10	10	10	10	10	9	-	-	-	-
Cow BW ^2^	435.2	444.1	470.2 ^a^	497.4 ^b^	567.7	557.9	4.66	<0.01	0.17	<0.01
Cow BCS ^3^	5.38	5.45	5.43 ^a^	5.65 ^b^	5.38	5.28	0.06	<0.01	0.35	0.02
DMI kg/d	12.86	12.58	15.15	17.07	16.59	16.59	1.00	<0.01	0.77	0.21
DMI g/kg of BW	29.57	28.24	26.63	28.69	29.35	30.22	1.68	0.57	0.48	0.43
DMI g/min	92.24	92.60	145.88	144.83	135.04	132.45	7.95	<0.01	0.97	0.98
% CV ^4^	15.95	22.41	22.50	17.76	16.53	20.62	3.37	0.62	0.16	0.21
Time at feeder min/d	144.65	139.92	105.39	123.12	131.69	132.28	9.51	0.45	0.67	0.44

^1^ Pooled standard error of the means; ^2^ cow body weight (kg) at initiation of trial; ^3^ cow body condition score at initiation of trial; ^4^ coefficient of variation for intake expressed as kg/d; * means within row and cow age lacking common superscript differ (*p* < 0.05).

**Table 3 animals-11-01822-t003:** The influence of heifer post-weaning RFI on subsequent beef cow performance and dry matter intake behavior of 3 age classes of lactating, non-pregnant beef cows average 60-d post-partum (Study 2).

Category	Cow Age, Years						-	*p*-Value		
	2/3		5/6		8/9					
	Low RFI	High RFI	Low RFI	High RFI	Low RFI	High RFI	SE ^1^	Age	RFI	Age × RFI
Cows *n*	7	10	9	10	9	9	-	-	-	-
Cow BW ^2^	397.5	408.4	541.8 ^a^	476.7 ^b^	534.2 ^a^	516.8 ^b^	4.96	<0.01	0.12	<0.01
Cow BCS ^3^	4.22	4.19	4.75 ^a^	4.95 ^b^	4.72 ^a^	4.50 ^b^	0.05	<0.01	0.69	<0.01
Calf *n*	6	10	9	10	9	9	-	-	-	-
Calf BW ^4^	97.3 ^a^	91.9 ^b^	95.1	97.9	104.9	101.8	1.55	<0.01	<0.02	<0.03
Calf Julian birth date	70.2	66.8	75.5	73.4	76.1	76.6	1.41	<0.01	0.09	0.38
DMI kg/d	18.22	18.41	22.90	24.00	23.84	23.10	1.00	<0.01	0.88	0.57
DMI g/kg of BW	45.74	44.88	42.48	41.71	45.00	45.19	1.87	0.32	0.74	0.95
DMI g/min	127.70	123.71	166.33	162.48	168.58	163.12	11.58	0.02	0.81	0.99
% CV ^5^	13.25	12.91	9.01	11.17	12.54	11.49	1.46	0.68	0.88	0.45
Time at feeder min/d	149.42	154.30	140.00	153.07	149.28	152.41	11.99	0.99	0.75	0.88
Milk production kg	3.89 ^a^	2.77 ^b^	4.88 ^a^	4.22 ^b^	4.23	4.28	0.16	<0.01	<0.01	<0.01
Milk production g/kg of BW	9.76 ^a^	6.82 ^b^	9.17 ^a^	7.32 ^b^	8.04	8.42	0.38	<0.01	<0.01	<0.01

^1^ Pooled standard error of the means; ^2^ cow body weight (kg) at initiation of trial; ^3^ cow body condition score at initiation of trial; ^4^ calf body weight (kg) at weigh-suckle-weigh; ^5^ coefficient of variation for intake expressed as kg/d; * means within row and cow age lacking common superscript differ (*p* < 0.05).

## Data Availability

The data presented in this study are available in Appendix A Appendix A.

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
