# Peer review of "Influence of Residual Feed Intake and Cow Age on Dry Matter Intake Post-Weaning and Peak Lactation of Black Angus Cows"

_animals, 2021, doi:10.3390/ani11061822_

Round 1
Reviewer 1 Report
The present study aimed to investigate the effect of residual feed intake, as a maker of cow efficiency, on the performance of lactating and non-lactating beef cows. The authors focused on RFI in postweaning period of heifers and observed its lasting effect on their consequent performance and productivity, although the effect of cow ages was more potent. The results show some of interesting findings such as the effect of postweaning RFI on cow BW and milk production, which will be valuable for the future studies in this field. This reviewer has some minor comments to address as indicated below;
- The experimental condition should be described with more clarity;
- The breeding schedule at the research farm should be mentioned.
- The timing of weaning in study 1 should be mentioned.
- The timing of calving period in study 2 should be mentioned.
- The classification of cow age are not consistent in Sup. Table 2. Please check.
- Tables; The items of RFI and P-value should be written in a single row. Superscripts, “a” should be indicated with higher values.
- Statistics; Posthoc test seems to be performed when interaction was detected between Age and RFI. The caption in tables says “Means within rows lacking…” but Low RFI and High RFI seems to be compared with in each age group. Please clarify.
- L157; The sentence does not match the result shown in Table 3.
- L162-165; This sentence is about the result of calves’ performance. The sentence from L166 – 170 is also about calves. Please think of combining these two to make one paragraph.
- Reference; Follow the reference style of this journal.
Author Response
Response to Animals Reviewer #1
- The experimental condition should be described with more clarity;
- The breeding schedule at the research farm should be mentioned.
- The timing of weaning in study 1 should be mentioned.
- The timing of calving period in study 2 should be mentioned.
Accepted reviewers’ suggestions and added:
All cattle were synchronized and time artificially inseminated (AI) corresponding to initial calving date of March 15, with exposure to cleanup bulls occurring approximately 7 days post AI for an additional 45 days. Calves were strategically weaned mid-September to mid-October each year with the timing based on fall forage conditions and/or cow body condition.
All cows were managed as one contemporary group post-weaning in the fall to calving in the spring of each year. All heifers/cows were exposed to electronic feed bunks as heifers and again during winter supplement studies.
- The classification of cow age are not consistent in Sup. Table 2. Please check.
Accepted reviewers’ suggestions and corrected cow age classifications in manuscript as well as supplementary tables.
- Tables; The items of RFI and P-value should be written in a single row. Superscripts, “a” should be indicated with higher values.
Accepted reviewers’ suggestions and placed RFI and P-values on same line. We also added a sentence in the materials and methods statistical section to clarify how we handled RFI by treatment interactions.
- Statistics; Posthoc test seems to be performed when interaction was detected between Age and RFI. The caption in tables says “Means within rows lacking…” but Low RFI and High RFI seems to be compared with in each age group. Please clarify.
Accepted reviewers’ suggestions, please see response to question #3 above.
- L157; The sentence does not match the result shown in Table 3.
Accepted suggestion and corrected text to match tabular data.
- L162-165; This sentence is about the result of calves’ performance. The sentence from L166 – 170 is also about calves. Please think of combining these two to make one paragraph.
Accepted suggestions and created one stand alone paragraph for all calf data.
- Reference; Follow the reference style of this journal.
Accepted suggestions and formatted all references.

Reviewer 2 Report
This manuscript addresses the important subject of feed efficiency and repeatability over different life-stages and stages of production. A lot more detail may help clarify some of the concerns mentioned below. My primary concerns are the short acclimation periods used, the very brief feed intake periods implemented, the data screening process applied to remove seven days of data (at least lack of any explanation or description of how this was done), lack of information about contemporary grouping and management prior to and during the experiment. My understanding is that the 2-year-olds were placed in the same pen with the older cows. This could possibly have a dramatic impact on study results, especially if they had not been managed together prior to the gestating and lactating studies.
39-40 Grazing plus purchased and harvested (supplemental nutrition?) feed costs combined generally cost 50 to 60% of annual expenses. https://www.agmanager.info/livestock-meat/production-economics/differences-between-high-medium-and-low-profit-cow-calf-1
94-95 Include description of the specific multiple regression equation(s) used predict feed intake and thus calculate RFI during the growing phase
102 A seven-day acclimation period does not seem adequate a) for animals that have not been exposed to the feed intake system for several years; b) to allow adjustment to the new diet...assuming they were not fed these pellets for several weeks prior to the initiation of the study.
103-104 Please describe this data screening process in detail. What exactly is "seven of the highest accuracy days of DMI recordings"? If half of the data (the most variable) were excluded, it is no wonder the pooled SE for feed intake is extremely low at 0.1 kg.
106 Provide details for management and contemporary grouping prior to the beginning of the experiment. Were these animals managed together for a long period of time prior to introducing them to the feed intake facility?
114 Table 1. Are you sure the chemical composition data is not switched? The hay/straw diet is 16.8% protein, for example?
118-119 At what stage of production/days post-calving was this experiment conducted?
122-123 If you are going to test the influence of treatment on calving date, provide details about the breeding system for each age group. Again, were all age groups managed as a contemporary group during the breeding season? All age groups exposed to AI, natural service equally? On the same day? Natural services bulls turned out at the same time, etc.? By restricting selection of animals to the first 42 days of the calving period, it is highly unlikely that treatment effects would be detected with only 7 - 10 animals in each treatment cell and only 17-20 in each age group. Effects seen in Table 3 for birth date could be/are likely the result of management rather than influence of age or heifer RFI.
126 Here, a seven-day feed intake period is indicated. Is it a seven-day period or a 14-day period, screened down to seven days of data?
126-128 Milk yield is extremely low according to Table 3; perhaps this was done during late-lactation? Please provide details related to the WSW procedure (beyond referencing Williams) so the reader can determine if the procedures may have influenced your results.
130-139 Nutrient requirements and feed intake can be substantially influenced by stage of gestation and stage of lactation. The statistical models do not appear to account for this potential source of variation. Please clarify.
151 Clarify which "intake" metric this CV applies to
152 Use "affected"
Table 2 and Table 3. Clarify what metric the "% CV" applies to. If this CV is associated with one of the feed intake metrics, I am not sure it is relevant. According to my understanding, the data were restricted / screened in some fashion which would have a major impact on this value. More information about the rules used to screen the data is required.
Author Response
Response to Animals Reviewer #2
This manuscript addresses the important subject of feed efficiency and repeatability over different life-stages and stages of production. A lot more detail may help clarify some of the concerns mentioned below. My primary concerns are the short acclimation periods used, the very brief feed intake periods implemented, the data screening process applied to remove seven days of data (at least lack of any explanation or description of how this was done), lack of information about contemporary grouping and management prior to and during the experiment. My understanding is that the 2-year-olds were placed in the same pen with the older cows. This could possibly have a dramatic impact on study results, especially if they had not been managed together prior to the gestating and lactating studies.
Accepted suggestions. See responses to reviewer #1 and below responses to reviewer #2 suggestions.
39-40 Grazing plus purchased and harvested (supplemental nutrition?) feed costs combined generally cost 50 to 60% of annual expenses. https://www.agmanager.info/livestock-meat/production-economics/differences-between-high-medium-and-low-profit-cow-calf-1
The authors appreciate the reviewers’ comment. The references used for introduction suggest this number may be as high as 65%. The actual amount of supplemental feed cost likely varies by location, climate and ranch resources.
94-95 Include description of the specific multiple regression equation(s) used predict feed intake and thus calculate RFI during the growing phase
Accepted suggestions and added the following:
Individual heifer post-weaning RFI was calculated following parameters set forth by Archer et al. [22], and Arthur et al. [4]:
RFIPhe = FI-β w(phe)×MWT-βg(phe)×DG,
where RFIphe = phenotypic residual feed intake, FI = daily feed intake, MWT = metabolic body weight at mid-test, DG = average daily gain, and βw(phe) and βg(phe) = partial regression coefficients of animal’s FI on MWT and DG, respectively.
102 A seven-day acclimation period does not seem adequate a) for animals that have not been exposed to the feed intake system for several years; b) to allow adjustment to the new diet...assuming they were not fed these pellets for several weeks prior to the initiation of the study.
Accepted suggestions and added clarification in the materials and methods. Specifically, we added the following:
All cows were managed as one contemporary group post-weaning in the fall to calving in the spring of each year. All heifers/cows were exposed to electronic feed bunks as heifers and again during winter supplement studies.
Individual animal intake was continuously recorded. The system was monitored daily for unaccounted feed balance. When greater than 5% of the feed disappearance was unaccounted for, the GrowSafe system automatically deemed the 24-h period as failed. Therefore, we selected the last 7 days of the 21-d period that met the criteria to calculate average DMI per individual animal for each DMI study. Variation in dry matter intake, measured as the coefficient of variation (% CV), was based on daily intake estimates for individual animals.
103-104 Please describe this data screening process in detail. What exactly is "seven of the highest accuracy days of DMI recordings"? If half of the data (the most variable) were excluded, it is no wonder the pooled SE for feed intake is extremely low at 0.1 kg.
Accepted suggestions. Please see above response.
106 Provide details for management and contemporary grouping prior to the beginning of the experiment. Were these animals managed together for a long period of time prior to introducing them to the feed intake facility?
Accepted suggestions. Please see above response.
114 Table 1. Are you sure the chemical composition data is not switched? The hay/straw diet is 16.8% protein, for example?
Accepted suggestions. The authors appreciate the reviewers’ attention to details.
118-119 At what stage of production/days post-calving was this experiment conducted?
Accepted suggestions. Study 1 was conducted on dry, non-lactating cows 14 days post weaning (approximately 200 days post calving). We added content in the beginning of our materials and methods to provide clarification for the reader.
122-123 If you are going to test the influence of treatment on calving date, provide details about the breeding system for each age group. Again, were all age groups managed as a contemporary group during the breeding season? All age groups exposed to AI, natural service equally? On the same day? Natural services bulls turned out at the same time, etc.? By restricting selection of animals to the first 42 days of the calving period, it is highly unlikely that treatment effects would be detected with only 7 - 10 animals in each treatment cell and only 17-20 in each age group. Effects seen in Table 3 for birth date could be/are likely the result of management rather than influence of age or heifer RFI.
Accepted suggestions. We attempted to group cows, so they were in the similar stage of lactation. In a companion study, we evaluated cow performance and reproductive performance for a 5 year production period as influenced by residual feed intake. We saw minimal differences in cow birth dates, cow body weight and cow body condition over 3 calf crops and breeding of the 4th.
Parsons, C.T.; Dafoe, J.M.; Wyffels, S.A.; DelCurto, T.; Boss, D.L. 2021. Impacts of heifer postweaning residual feed intake classification on reproductive and performance measurements of first, second, and third parity Angus beef females. Transl. Anim. Sci. doi:10.1093/tas/txab061.
126 Here, a seven-day feed intake period is indicated. Is it a seven-day period or a 14-day period, screened down to seven days of data?
Accepted suggestions. See above comments related to line 102.
126-128 Milk yield is extremely low according to Table 3; perhaps this was done during late-lactation? Please provide details related to the WSW procedure (beyond referencing Williams) so the reader can determine if the procedures may have influenced your results.
Accepted suggestions. We added the following sentence.
For our study, we utilized an 8-hr calf removal protocol.
For rangeland beef cattle production, the milk production levels are typical for cattle fitted to a limited nutritional environment. Although our methods are different, these values are similar to those obtained by others.
Grings, E. E., Roberts, A. J., Geary, T. W., & MacNeil, M. D. (2008). Milk yield of primiparous beef cows from three calving systems and varied weaning ages. Journal of Animal Science, 86(3), 768-779.
130-139 Nutrient requirements and feed intake can be substantially influenced by stage of gestation and stage of lactation. The statistical models do not appear to account for this potential source of variation. Please clarify.
Accepted suggestions.
151 Clarify which "intake" metric this CV applies to
Accepted suggestions. Added the following sentence.
Variation in dry matter intake, measured as the coefficient of variation (% CV), was based on daily intake estimates for individual animals.
152 Use "affected"
Accepted suggestions.
Table 2 and Table 3. Clarify what metric the "% CV" applies to. If this CV is associated with one of the feed intake metrics, I am not sure it is relevant. According to my understanding, the data were restricted / screened in some fashion which would have a major impact on this value. More information about the rules used to screen the data is required.
Accepted suggestions. An explanation of how % CV was calculated was added in the materials and methods section. See above.

Round 2
Reviewer 2 Report
The explanation of the screening process is helpful. The authors have improved methods description considerably. A few additional considerations:
120 Indicate the CV relates to DMI, kg/d if that is correct. Tables should stand alone. Clearly indicate that the row of data for %CV relates to DMI, kg/d if that is correct. This row of data is directly beneath the row of data representing DMI, g/minute so it is confusing as presented.
127 Shouldn't the youngest age group be 2/3 instead of 1/2? If initiation of trial is 14-d post weaning, surely there are no cows that have not yet reached 2 years of age?
134-146 What stage of lactation was the 7-day intake period and the WSW procedure conducted?
Again, something seems amiss as pooled SE for DMI, kg/d in Table 2 = 0.1. However, % CV (assuming it relates to DMI, kg/d) is around 20%. In Table 3, pooled SE for DMI = 1.0 whereas % CV is around 11 to 12%.
Author Response
Reviewer #2
The explanation of the screening process is helpful. The authors have improved methods description considerably. A few additional considerations:
120 Indicate the CV relates to DMI, kg/d if that is correct. Tables should stand alone. Clearly indicate that the row of data for %CV relates to DMI, kg/d if that is correct. This row of data is directly beneath the row of data representing DMI, g/minute so it is confusing as presented.
This % CV is related to kg ∙ d-1 as you noted. We also added an additional superscript
4Coefficient of variation for intake expressed as kg ∙ d-1 at the bottom of tables 2 and 3.
127 Shouldn't the youngest age group be 2/3 instead of 1/2? If initiation of trial is 14-d post weaning, surely there are no cows that have not yet reached 2 years of age?
See comments in manuscript. These 1/2-year-old cows are actually coming 2/3-year old cows. The 1-year old cows are actually 18 months at time of the feeding trial and the 2-year-old cows are actually coming 3, having just weaned their first calf. We realize this gets a bit confusing, but this is the best way we have found to explain these age classes.
134-146 What stage of lactation was the 7-day intake period and the WSW procedure conducted?
This weigh suckle weigh study was done when the calves averaged 60 days of age with an age range from 39 – 81 days of age.
Again, something seems amiss as pooled SE for DMI, kg/d in Table 2 = 0.1. However, %CV (assuming it relates to DMI, kg/d) is around 20%. In Table 3, pooled SE for DMI = 1.0 whereas % CV is around 11 to 12%.
Accepted suggestions. The authors would like to thank you for your observation of the 0.10 SE of the DMI kg ∙ d-1. We went back through our stats and found that the actual SE is 1.0 for this variable and have corrected in table 2.
